# Mitochondrial Metal Ion Transport in Cell Metabolism and Disease

**DOI:** 10.3390/ijms22147525

**Published:** 2021-07-14

**Authors:** Xuan Wang, Peng An, Zhenglong Gu, Yongting Luo, Junjie Luo

**Affiliations:** 1Beijing Advanced Innovation Center for Food Nutrition and Human Health, Key Laboratory of Precision Nutrition and Food Quality, Department of Nutrition and Health, China Agricultural University, Beijing 100193, China; xuanwxuan@outlook.com (X.W.); an-peng@cau.edu.cn (P.A.); 2Division of Nutritional Sciences, Cornell University, Ithaca, NY 14853, USA; zg27@Cornell.edu

**Keywords:** mitochondrial metal ion transport, mitochondrial metal ion homeostasis, mitochondrial function, cell metabolism, disease

## Abstract

Mitochondria are vital to life and provide biological energy for other organelles and cell physiological processes. On the mitochondrial double layer membrane, there are a variety of channels and transporters to transport different metal ions, such as Ca^2+^, K^+^, Na^+^, Mg^2+^, Zn^2+^ and Fe^2+^/Fe^3+^. Emerging evidence in recent years has shown that the metal ion transport is essential for mitochondrial function and cellular metabolism, including oxidative phosphorylation (OXPHOS), ATP production, mitochondrial integrity, mitochondrial volume, enzyme activity, signal transduction, proliferation and apoptosis. The homeostasis of mitochondrial metal ions plays an important role in maintaining mitochondria and cell functions and regulating multiple diseases. In particular, channels and transporters for transporting mitochondrial metal ions are very critical, which can be used as potential targets to treat neurodegeneration, cardiovascular diseases, cancer, diabetes and other metabolic diseases. This review summarizes the current research on several types of mitochondrial metal ion channels/transporters and their functions in cell metabolism and diseases, providing strong evidence and therapeutic strategies for further insights into related diseases.

## 1. Introduction

Mitochondria are cytoplasmic organelles crucial to life. Since the major function of mitochondria is to produce a large amount of energy ATP, they are called the powerhouse of the cell. However, the functions of mitochondria are multifaceted, far beyond bioenergetics, such as cell metabolism, apoptosis, fatty acid *β*-oxidation, reactive oxygen species (ROS) signaling, steroid synthesis, and metal ion homeostasis [1,2,3,4,5]. Mitochondria have a double membrane structure, namely mitochondrial inner membrane (MIM) and mitochondrial outer membrane (MOM), both of which contain selective and nonselective ion channels and transporters [6,7]. The MOM with a simple structure is a permeable membrane for small molecules and ions, while the MIM is highly impermeable and its passive transport mode is usually driven by electrochemistry [8,9]. The recently discovered mitochondrial metal ion channels/transporters are mainly located on these mitochondrial membranes, especially the MIM. While some metal ion channels/transporters participate in normal physiological activities, others only play their roles in pathological states, both of which are essential for cell survival and metabolism. Moreover, mitochondrial metal ion channels/transporters are considered to be important communication media between mitochondria and cytoplasm. Unbalanced communication causes disorders of cell metabolism and energy supply, leading to multiple pathologies [10,11].

Impaired metal ion homeostasis at the cellular level is linked to mitochondrial dysfunction [12,13]. The dynamic balance of metal ions inside and outside the mitochondria plays significant roles in numerous cellular physiological processes, including activating ATPase, maintaining ATP production, keeping the homeostasis of mitochondrial volume, regulating the concentration of ROS, controlling signal transduction, and holding the balance of other ion concentrations. Furthermore, it has been reported that the metal ion dyshomeostasis in mitochondria is related to pathological features in neurodegenerative diseases, such as Alzheimer’s disease and Parkinson’s disease [12,14,15]. In addition, diseases related to mitochondrial metal ion dyshomeostasis also include cancer, type 2 diabetes (T2D), heart failure, ischemia and reperfusion injury [16,17,18,19,20]. It is particularly important to note that targeting certain mitochondrial metal ion channels/transporters can treat diseases, such as hypoxic pulmonary artery hypertension, cancer and neurodegenerative disorders [21,22,23].

As far as we know, although the current research on mitochondrial metal ion transport is still limited, it has been found that there are several specific channels/transporters on the inner or outer membrane of mitochondria, which are responsible for transporting different metal cations, such as Ca^2+^, K^+^, Na^+^, Mg^2+^, Zn^2+^ and Fe^2+^/Fe^3+^ (Table 1). Mitochondrial channels/transporters and transports of metal cations are the key to modulating metal ion homeostasis directly or indirectly, which is essential for mitochondrial function, cellular metabolism, health and disease (Figure 1). Herein, our review summarizes the current relevant studies and focuses primarily on several types of mitochondrial metal ion transport and their roles in cell metabolism and diseases. Not only can it provide a reference for in-depth research on the transport of mitochondrial metal ions, but is also expected to develop more disease treatment strategies.

**Table 1 ijms-22-07525-t001:** Different mitochondrial channels/transporters transport various metal ions, which are closely related to many types of diseases. Most of the discovered mitochondrial metal ion channels/transporters are located on the MIM, but there are a few exceptions, such as VDAC and DMT1 (on the MOM).

Metal Ions	Mitochondrial Channels/Transporters		Related Diseases	References
	Importer/Influx	Exporter/Efflux		
Ca^2+^	VDAC, MCU, mRYR	Letm1, NCLX, mPTP	Insulin resistance, T2D, Diabetes-related cardiac disease, Heart failure, Ischemia, Reperfusion injury, Brain aging, Neurodegenerative diseases, Cancer	[17,20,24,25,26,27,28,29,30,31,32,33,34,35,36]
K^+^	mitoKATP, KCa, Kv, mitoTASK-3	KHE	Epilepsy, Diabetic cardiomyopathy, Ischemia, Reperfusion injury, Pulmonary artery hypertension, Neurodegeneration, Cancer, Schizophrenia, Sudden cardiac death	[37,38,39,40,41,42,43,44,45,46,47,48,49,50,51,52,53]
Na^+^	NCLX	NHE	Heart failure, Sudden death, Neurodegenerative diseases	[18,35,54,55,56,57]
Mg^2+^	MRS2	SLC41A3, Mme1	Cancer, Demyelination, Neurodegeneration	[58,59,60,61,62,63,64,65]
Zn^2+^	MCU, ZnT4	ZIP8, mitoKATP	Neurodegeneration	[66,67,68,69,70,71]
Fe^2+^/Fe^3+^	MFRN, Tf/TfR2, DMT1	——	Anemia, Neurodegenerative diseases	[72,73,74,75,76,77,78,79,80,81]

## 2. Mitochondrial Ca^2+^

Mitochondria produce ATP through OXPHOS to provide energy for various physiological activities in cells, and the Ca^2+^ concentration in mitochondria dynamically regulates the rate of ATP production [82]. Mitochondria link Ca^2+^ transport processes to the cellular metabolic state and affect the entire cellular network of Ca^2+^ signaling [83]. Mitochondrial Ca^2+^ modulates a number of Ca^2+^-dependent proteins and enzymes, to which many cellular activities and physiological and pathophysiological processes are related [17,84,85].

The mitochondrial Ca^2+^ influx is mainly mediated by the voltage-dependent anion-selective channel (VDAC), mitochondrial calcium uniporter (MCU) and mitochondrial ryanodine receptor (mRYR) transporter. The VDAC is a member of the *β*-barrel membrane protein family, which is ubiquitous on the MOM of mammals [86,87]. It allows Ca^2+^ and other small molecular solutes in and out and is essential for Ca^2+^ transport between the cytoplasm and the mitochondrial membrane space [24,88]. However, the control and regulation of mitochondrial Ca^2+^ concentration is primarily on the MIM. The MCU is a mitochondrial calcium selective channel that regulates the balance of Ca^2+^ concentration between the cytoplasm and the mitochondrial matrix, thereby preventing excessive or insufficient calcium intake into the mitochondria [25,89,90]. In order to prevent detrimental Ca^2+^ overload, the activity of MCU must be tightly regulated [91]. The MCU-mediated Ca^2+^ uptake is driven by the inner membrane potential produced by the electron-transport chain (ETC) [92]. MCU channels are closed when Ca^2+^ is in equilibrium and activated when the concentration of Ca^2+^ in cytoplasm elevates [93,94]. Interestingly, the mRYR is also involved in mitochondrial Ca^2+^ influx, the dysregulation of which causes the onset of mitochondrial retrograde signaling in mouse and human cell lines [26,95].

The mitochondrial Ca^2+^ efflux pathways mainly include mitochondrial Ca^2+^/H^+^ antiporter, Na^+^/Ca^2+^ exchanger and mitochondrial permeability transition pore (mPTP). Letm1, a mitochondrial inner membrane protein, has been identified as a mitochondrial Ca^2+^/H^+^ antiporter, which is essential for normal glucose metabolism and alters brain function in Wolf–Hirschhorn syndrome mouse models [27,96,97]. The NCLX is an essential component of mitochondrial Na^+^/Ca^2+^ exchanger, which plays a fundamental role in regulating mitochondrial Ca^2+^ homeostasis [28,98]. The mPTP is a transmembrane protein residing on the MIM which can be induced by excessive Ca^2+^ in the mitochondrial matrix [29,99]. In turn, the mPTP leads to the collapse of the mitochondrial transmembrane potential and induces Ca^2+^ to release nonspecifically from the mitochondrial matrix to the cytoplasm, which is called “calcium-induced calcium release”. Thus, the mitochondrial Ca^2+^ uptake should be strictly regulated to maintain a low matrix Ca^2+^ concentration that meets the dynamic cellular energy requirements and prevents mPTP from opening [92].

Mitochondrial Ca^2+^ transport participates in OXPHOS, tricarboxylic acid cycle, fatty acid metabolism and apoptosis related to the pathogenesis of some diseases [17,100,101,102,103]. Generally, the influx and efflux rates of Ca^2+^ in mitochondria are in balance, while the imbalance of mitochondrial Ca^2+^ homeostasis results in a series of changes in the physiological state of cells and diseases. For example, a study has indicated the crosstalk between mitochondrial Ca^2+^ uptake and autophagy in skeletal muscle [104]. Excessive Ca^2+^ within mitochondria can induce apoptosis by mPTP [105]. The reduced mitochondrial Ca^2+^ overload under pro-apoptotic stimuli significantly alleviated the apoptotic response [106]. Mitochondrial Ca^2+^ is also involved in insulin signaling [107]. The dysregulation of mitochondrial Ca^2+^ influx and efflux can disrupt intracellular Ca^2+^ homeostasis, which leads to insulin resistance and T2D in humans and animals [17]. By regulating mitochondrial calcium handling can alleviate diabetes-related cardiac disease [30]. It has been reported that insufficient mitochondrial Ca^2+^ uptake or Ca^2+^ overload contributes to heart failure, ischemia, reperfusion injury, brain aging, and neurodegenerative disease (e.g., Huntington’s disease, Parkinson’s disease and Alzheimer’s disease) [20,31,32,33,34,35,108]. Targeting mitochondrial calcium has been considered a potential treatment against Parkinson’s disease [109]. The loss of NCLX is a new driver of metastasis, and the regulation of mitochondrial Ca^2+^ is a novel therapeutic approach in human metastatic colorectal cancer [36]. Hopefully, mitochondrial Ca^2+^ transport will become potentially attractive targets for the development of novel therapeutic strategies.

## 3. Mitochondrial K^+^

A very significant role of K^+^ in mitochondria is to maintain volume homeostasis and prevent excessive swelling or contraction of the mitochondrial matrix [110]. The intramitochondrial K^+^ concentration is greater than the cytosolic K^+^ concentration in neurons, suggesting that there must be K^+^ channels/transporters between the mitochondria and cytoplasm [111]. Not surprisingly, it has been discovered that mitochondrial K^+^ transport channels including mitochondrial ATP-sensitive K^+^ (mitoKATP) channels, Ca^2+^-activated K^+^ (KCa) channels, voltage-gated K^+^ (Kv) channels, and TWIK-related acid-sensitive K^+^ channel 3 (mitoTASK-3) are located on the MIM [37,38,39,40,41]. The functions of these K^+^ channels involve regulating mitochondrial respiration, maintaining membrane potential, and producing ROS [112]. Enhanced mitochondrial K^+^ influx can induce cardioprotection and also protect against different forms of cell damage and death in different tissues [113,114,115].

The MitoKATP channel, as a highly selective conductor of K^+^, is an important factor in controlling mitochondrial and cellular physiology. Abundant MitoKATP channels respond to the cellular energetic status by regulating organelle volume and function [116]. MitoKATP channels have potential effects on multiple pathological processes, such as improving cardiac function, inhibiting apoptosis in diabetic cardiomyopathy, preventing ischemia and brain reperfusion injury, and protecting against partial rhythm disorders [44,45,46,47]. The protective role of MitoKATP channels is also found in temporal lobe epilepsy through the regulation of mitochondrial dynamic proteins in a rat model [43]. In addition, MitoKATP channels promote the proliferation of hypoxic human pulmonary artery smooth muscle cells and have an important role in the development of hypoxic pulmonary artery hypertension in animal models, which may become a target for disease treatment [23,48].

KCa channels open at the physiological inner membrane potential depending on the concentration of Ca^2+^ and selectively transport K^+^ [117]. In the cardiovascular system, KCa channels are considered to be key mediators that control vascular tone and blood pressure by modulating the membrane potential and shaping Ca^2+^-dependent contraction [118]. The KCa channel opener NS-1619 has been shown to prevent the heart from being affected by global ischemia and reperfusion in mice [119,120]. In addition, KCa channels have emerged as a potential therapeutic tool for aging and age-related neurodegeneration [49]. Kv channels involve a variety of physiological processes ranging from triggering neuronal and cardiac action potential to regulating cell cycle and cell volume, to driving cellular proliferation and migration [16,50]. These Kv channels have a major role in human diseases such as cancer, schizophrenia, epilepsy, and sudden cardiac death [50,51,53]. In recent years, emerging evidence demonstrates that the Kv channel is becoming an oncological target since targeting it can reduce tumor growth and progression in vitro and in vivo [52,121,122,123]. Another K^+^ influx channel, mitoTASK-3, whose function may be involved in ROS production and apoptosis, needs to be further understood [124,125]. Notably, recent research results point out that mitoTASK-3 has an effect on mitochondrial physiology, cancer cell survival and migration, and is a potential therapeutic target [126].

In turn, the K^+^ efflux is conducted through a mitochondrial K^+^/H^+^ exchanger (KHE), which is activated with the expansion of mitochondrial volume, preventing excess matrix swelling [42,127]. The defect of KHE can lead to increased matrix K^+^ content, swelling, and attenuation of autophagy [128]. Compared with K^+^ influx channels/transporters, the number of mitochondrial K^+^ efflux channels/transporters that have been reported is extremely limited (Table 1), which is not conducive to the study of mitochondrial K^+^ homeostasis and related diseases and requires more attention.

## 4. Mitochondrial Na^+^

A recent study has revealed that Na^+^ controls OXPHOS function and redox signal transduction through unexpected interactions with phospholipids and has a profound impact on cell metabolism [129]. Mitochondria in living cells keep low Na^+^ levels despite the large electrochemical gradient favoring cation influx into the matrix, suggesting a complex transport mechanism is involved [130]. There is already evidence that mitochondrial Na^+^ transients are governed by the mitochondrial Na^+^/H^+^ exchanger (NHE), which mediates mitochondrial Na^+^ efflux [56]. The Na^+^ spiking activity is significantly inhibited by mitochondrial NHE inhibition and is sensitive to cellular pH and Na^+^ concentrations [130]. On the contrary, the Na^+^ influx into the mitochondria is primarily mediated by the mitochondrial Na^+^/Ca^2+^ exchanger (NCLX) [54,55]. An elevated Na^+^ concentration promotes mitochondrial Ca^2+^ efflux via NCLX [131]. In fact, mitochondrial Na^+^ influx through NCLX could be potentially counter-balanced by the activity of a mitochondrial NHE. The NCLX is essential for cardiac survival and knockout of NCLX leads to heart failure in mice [18]. Another study shows that inhibition of the Na^+^/Ca^2+^ exchanger can prevent sudden death in a guinea pig model [57]. In addition, the mitochondrial NCLX, as a novel therapeutic target, also plays a role in a variety of neurodegenerative disorders [35].

## 5. Mitochondrial Mg^2+^

Mitochondria are believed to be responsible for cellular Mg^2+^ homeostasis [64]. The metabolism and productivity of mitochondria may lead to dynamic changes in Mg^2+^, which plays a vital role in the respiratory system [132,133]. Not only can mitochondria accumulate Mg^2+^, but also release Mg^2+^. Changes in mitochondrial Mg^2+^ concentration affect Mg^2+^-sensitive matrix enzymes (e.g., pyruvate dehydrogenase) and transporters (e.g., Ca^2+^ uniporter) on the MIM [134]. The Mg^2+^ concentration, remarkably constant and low in cytoplasm but tenfold higher in mitochondria, mediates ADP/ATP exchange between the cytosol and the matrix [135]. Mg^2+^ and ATP form a stable Mg–ATP complex, and the degree of complex formation depends on the ratio of Mg^2+^ to ATP [136]. Since Mg–ATP is in a balanced state with free Mg^2+^ in cells, it has been suggested that a decrease in free Mg^2+^ may be a primary cause of low intracellular ATP [137]. The Mg^2+^ concentration needs to be adjusted to equilibrate any changes in rapidly available free energy [138]. Mg^2+^ participates in the release of Cytochrome C, and the concentration of Mg^2+^ in the cytoplasm of apoptotic cells is also significantly increased in vitro [139,140]. However, in human colon cancer cells, intracellular Mg^2+^ content decreases during mitochondria-mediated apoptosis [141]. Regardless of this controversy, it is certain that there is a close relationship between Mg^2+^ homeostasis and apoptosis. In short, the effects of Mg^2+^ on mitochondrial functions mainly focus on energy metabolism, mitochondrial Ca^2+^ processing and apoptosis [139].

It has been found that the Mg^2+^ transporter of MRS2 (mitochondrial RNA splicing protein 2) on the MIM is an essential component of the mitochondrial Mg^2+^ uptake system which can selectively transport Mg^2+^ [58,59,133]. The MRS2 channel seems to be involved in apoptosis, and its expression is related to the Mg^2+^ content inside the cells and mitochondria [142,143]. In fact, in vitro experiments conducted on human cells of conditional knockdown and overexpression of MRS2 channels induce cell death or lead to a minor susceptibility to apoptotic pharmacological insults [142,143]. Moreover, mitochondria-mediated apoptosis is linked to the multidrug-resistant (MDR) phenotype and gastric cancer, and the function of human MRS2 protein may be a promising target for MDR reversal therapy [65]. There is also a report showing that a mutation in the gene encoding mitochondrial Mg^2+^ channel MRS2 results in demyelination in rats [63]. Rats with MRS2 defect exhibit severe rapid myelin breakdown throughout the central nervous system [144]. Another two mitochondrial Mg^2+^ transporters, namely the mitochondrial Mg^2+^ efflux system SLC41A3 and the mitochondrial carrier protein Mme1, export Mg^2+^ into the cytoplasm and work together with mitochondrial importers to accurately regulate Mg^2+^ homeostasis [60,61,62]. In living cells, once mitochondrial Mg^2+^ homeostasis is out of balance, ATP production will be disrupted via a shift in mitochondrial energy metabolism and morphology [133]. Moreover, the perturbation of Mg^2+^ in cells and mitochondria has been shown to be involved in the neurodegenerative process of Parkinson’s disease in cell models [64].

## 6. Mitochondrial Zn^2+^

The Zn^2+^ level is the most abundant in mitochondria compared with in other subcellular compartments [145]. Zn^2+^ plays an important role in maintaining mitochondrial function, and the dynamic balance of Zn^2+^ in mitochondria is necessary for normal cell metabolism. Zinc is thought to be beneficial in protecting mitochondrial antioxidants and ETC enzymes [146]. Movement of Zn^2+^ between cytosolic and mitochondrial pools might be of functional significance in intact neurons [147]. Zn^2+^ accumulates in neurons after ischemia and induces mitochondrial dysfunction and cell death [148,149]. Overloading of cellular or intramitochondrial Zn^2+^ will lose mitochondrial membrane potential, enhance ROS production and reduce cellular ATP levels [150,151]. Therefore, it is very important to study Zn^2+^ transport between cytosol and mitochondria. Some studies have indicated that Zn^2+^ is transported into mitochondria through MCU and suggested that Zn^2+^ is a critical contributor to mitochondrial dysfunction and ischemic neurodegeneration [66,67,68]. In MCU knockout mice, mitochondrial Zn^2+^ accumulation is significantly reduced [66]. Ca^2+^ may markedly increase the permeability of MCU to Zn^2+^, which has a synergistic effect [152]. It is inferred that the co-regulatory proteins MICU1 and MICU2, regulating mitochondrial Ca^2+^, also indirectly affect Zn^2+^ transport by MCU. In addition, two studies propose that mitochondrial SLC30A4 (Zinc transporter 4, ZnT4) transports Zn^2+^ from the cytosol to mitochondria, while mitochondrial SLC39A8 (ZIP8) transports Zn^2+^ from mitochondria to the cytosol [69,70]. In a rat model of alcohol-induced hepatic zinc deficiency, both SLC30A4 and SLC39A8 expression levels are observed to increase. Another report suggests that mitoKATP may be the main gate for the release of increased free Zn^2+^ from within the mitochondria in PC12 cell line [71]. Nevertheless, the transport mechanism of mitochondrial Zn^2+^ still needs more supporting evidence.

## 7. Mitochondrial Iron Ion

Mitochondria are the main utilization sites of iron which are transported to the matrix to synthesize iron-sulfur clusters and heme [153]. The precise regulation of iron ions in mitochondria is essential for hemoglobin production, Fe-S cluster protein assembly and heme biosynthesis during red blood cell development [78,154,155]. It is conceivable that mitochondrial iron homeostasis is involved in various hematological diseases. Mitochondrial iron homeostasis and its dysfunctions have been found in sideroblastic anemia and neurodegenerative disorders such as Alzheimer’s disease, Parkinson’s disease, Huntington disease, and Friedreich’s ataxia [78,79,80,81]. Strikingly, iron overload is the primary cause of increased morbidity in thalassemia [156], but the current research on the relationship between mitochondrial iron homeostasis and thalassemia is very rare. In fact, abnormal cellular iron metabolism is largely affected by mitochondrial iron dyshomeostasis, which may lead to iron overload associated side effects. In turn, an iron loss induced by iron chelator triggers mitophagy [157].

MFRN (SLC25A37) belongs to the vertebrate mitochondrial solute carrier protein family, which transports various metabolites and cofactors on the MIM. Some researchers show that the MFRN is a carrier of iron ions into mitochondria, and mitochondria of MFRN mutants disrupt iron ion uptake, leading to severe hypochromic anemia and stagnant red blood cell maturation [72,73,74,75]. In mouse embryonic stem cells, the lack of MFRN causes fibroblasts to stop maturing and inhibits heme synthesis [72]. The disruption of yeast MFRN orthologs MRS3 and MRS4 leads to defects in iron metabolism and mitochondrial Fe-S cluster formation [158,159]. The transferrin/transferrin receptor 2 (Tf/TfR2) transport system has been reported to deliver transferrin-bound iron to mitochondria, which is disrupted in Parkinson’s disease [76]. Another divalent metal transporter called DMT1 on the MOM also transports iron ions [77]. Overexpression of DMT1 is observed to increase the mitochondrial uptake of iron ions driven by proton gradients. Some small solutes and metal ions enter the mitochondrial membrane space through VDAC, which may involve iron uptake. Further research shows that VDAC is one of DMT1 interacting partners, and DMT1–VDAC interactions mediate mitochondrial iron uptake in cells [160]. Interestingly, mitochondrial ferritin (FTMT), as a novel iron-storage protein in mitochondria, participates in regulating iron distribution between cytosol and mitochondrial contents and has a protective effect in pathogenesis of neurodegenerative diseases in cell models [161,162]. It is worth noting that mitochondrial iron ion efflux channels/transporters have not yet been discovered, which requires an in-depth investigation (Table 1).

## 8. Mitochondrial Manganese Ion

Mitochondria are the main organelles producing ROS, the accumulation of which will cause cell toxicity, accelerated mutagenesis, lipid peroxidation, protein oxidation and cell death. Thus, the primary mechanism to eliminate ROS relies on superoxide dismutase (SOD), including the cytosolic copper/zinc-dependent SOD (Cu/ZnSOD) and the mitochondrial manganese-dependent SOD (MnSOD) [163]. MnSOD is the only SOD isoform present in mitochondria, which requires a manganese ion as a cofactor to execute its antioxidant defense function [164]. At the cellular level, SLC39A8 (ZIP8) and SLC39A14 (ZIP14) have been identified to specifically mediate manganese uptake in mammals [165,166,167], while SLC30A10 (ZnT10) controls manganese efflux from cells [168,169]. However, the manganese ion transport mechanism in mitochondria is still unclear. It has been proposed that manganese ion uptake from cytosol to mitochondria is possibly mediated via MCU or MFRN1 [170]. Cells lacking MCU are more resistant to Mn^2+^ toxicity [171]. A potential role of mitochondria-localized SLC39A8 in the regulation of mitochondrial manganese ion transport will be an interesting point in future research. Of note, the potential role for altered Mn homeostasis and toxicity in neurodegenerative disorders has been reported, but whether it is related to mitochondrial manganese ion transport is still unknown [172,173,174].

## 9. Other Mitochondrial Metal Ions

Actually, there are more types of metal ions in mitochondria, not just those described above. With reference to the method of rapid immunopurification of mitochondria [175,176], we isolated mitochondria from HepG2 cells and measured the content of different mitochondrial metal ions by inductively coupled plasma mass spectrometry (ICP–MS). This ICP–MS method has been reported to provide high accuracy in monitoring various metal ions [177,178]. As shown in Figure 2, our results demonstrate that the seven most abundant metal elements in mitochondria are K, Ca, Na, Mg, Fe, Mn and Zn, whose mitochondrial transport channels and transporters have attracted great attention from researchers. So, their transport channels and transporters have been discovered more or less. However, Al, Cu, Sn, Ni, Ba and other metal ions whose abundances are closely followed and relatively low should also be considered seriously. On the one hand, what are the functions of these metal ions in mitochondria? Are they related to some mitochondrial diseases? On the other hand, how they are transported into and out of mitochondria? These questions have not yet been fully answered and require further investigation. For instance, aluminum is considered to be an inducer of the mitochondrial permeability transition [179], and aluminum phosphide can induce oxidative stress and mitochondrial damages in cardiomyocytes and isolated mitochondria [180]. Unfortunately, the transport of mitochondrial aluminum ions is still unclear. Another example is about mitochondrial copper ions. The study has found mitochondrial copper homeostasis and its derailment in Wilson disease [181]. Disruption of mitochondrial copper distribution inhibits self-renewal of leukemia stem cells [182]. More interestingly, it has recently been discovered that mitochondrial copper depletion can suppress triple-negative breast cancer [183]. These studies indicate that mitochondrial copper may be a potentially important and non-negligible target in therapy. However, mitochondrial copper ion transport is poorly understood. As for the lower contents of Sn, Ni, Ba and other metal ions, it is not yet clear how they are transported between the cytoplasm and the mitochondrial matrix. In brief, it is necessary to comprehensively investigate the transport of these mitochondrial metal ions in cell metabolism and diseases, which may become a potential target for the treatment of related diseases.

## 10. Concluding Remarks and Prospects

Under normal physiological conditions, the influx and efflux of mitochondrial metal ions are in dynamic equilibrium. Each channel/transporter transports metal ions at a very fast speed and generates huge electrochemical energy. In order to maintain the ATP content produced by the proton motive force, the opening probability of these metal ion channels and the number of metal ion channels/transporters of each mitochondrion must be tightly modulated to balance the influx and efflux of metal ions. Generally speaking, mitochondrial metal ion channels/transporters, with a significant role in energy supply, metabolic cycle, cell survival and death, are considered to be highly selective and strictly regulated. However, some metal ions in the matrix are not necessary for mitochondria, or their presence may be harmful, which are possible due to nonspecific transportation of certain metal ion channels/transporters.

As summarized in Table 1, there are different channels/transporters on the inner or outer membranes of mitochondria to transport various kinds of metal ions. The MCU can transport Ca^2+^ and Zn^2+^ from the cytoplasm to the mitochondrial matrix and may also participate in mitochondrial manganese ion uptake, indicating that MCU is not a specific metal ion transport channel. Interestingly, while NCLX mediates Na^+^ influx into mitochondria, it can also export Ca^2+^ from the mitochondrial matrix to the cytoplasm. Increasing Na^+^ level promotes mitochondrial Ca^2+^ efflux, which is an important mechanism to maintain the balance of Na^+^-Ca^2+^ in mitochondria. Therefore, if NCLX is mutated, it will cause the dyshomeostasis of both Na^+^ and Ca^2+^. Similarly, mitoKATP not only transports K^+^ into mitochondria, but also releases Zn^2+^ from the mitochondrial matrix into the cytoplasm. It is speculated that mitoKATP may regulate the balance of K^+^-Zn^2+^ in mitochondria. Surprisingly, no mitochondrial iron efflux channel/transporter has been discovered so far. Considering that most metal ions cannot be accumulated indefinitely in mitochondria, their corresponding efflux channels/transporters exist theoretically.

Actually, within mitochondria there may be other concentrations of metal ions (e.g., calcium) than in the cytoplasm, which is crucial for proper functioning of mitochondria as an organelle. Mitochondrial metal ions homeostasis plays an important role in health. Excessive metal ions mean dyshomeostasis, which affects cell metabolism and disease. Metal ions chelators have been reported to solve excess metal ions in cells, and are used in a range of applications. However, metal ion chelators applied to mitochondria are rare at present. Whether chelators acting on the cytoplasm can also regulate the homeostasis of mitochondrial metal ions remains to be further studied. The development of potential chelators that act on mitochondrial metal ions will be challenging and of great significance.

As is well known, mitochondria are essential for the metabolism of cells, which is closely related to disease and health. Abnormal cell metabolism, apoptosis or cell death caused by mitochondrial dysfunction can lead to serious and terrible adverse consequences. Since metal ion channels/transporters play important roles in keeping the normal functions of mitochondria and cells, if there is a problem in metal ion transportation, resulting in metal ions dyshomeostasis, it will cause a series of unhealthy cellular physiological states and numerous related diseases. Among related diseases caused by the dyshomeostasis of metal ions, neurodegeneration and cardiovascular diseases have been observed to be the most common (Table 1). We speculate this for at least two reasons: (1) Due to the high incidence and harmfulness of these two diseases, they have received more attention from scientific researchers. (2) The brain and heart need more energy and have a higher mitochondrial abundance than other organs. Once the mitochondrial function is impaired, the brain and heart are more prone to disease. Neurodegenerative diseases, in particular, are associated with most mitochondrial metal ions, suggesting the importance of exploring specific causes and targeted treatments.

Fortunately, mitochondrial metal ion transport is becoming an effective therapeutic target for some diseases such as T2D and cancer. However, the translational research of more channels/transporters in health and disease still has a long way to go. The current research on mitochondrial metal ion channels/transportations is insufficient, and the transport mechanism is not fully revealed. It should be emphasized that, as shown in Figure 2, there are more metal ions with relatively low abundance in mitochondria, but very little is known about their transports and functions. In different types of cells, the abundance of mitochondrial metal ions may also be different. Therefore, it is necessary to further explore the fields of mitochondrial metal ion channels/transports and provide a more detailed theoretical basis for the development of new strategies or drugs on treating related diseases. Up to now, at least 1000 nuclear-encoded genes located in mitochondria have been found, which definitely include all mitochondrial channel/transporter genes. More new discoveries about metal ion transports and related diseases could be expected through the large-scale knockout/knockdown or overexpression of these genes.

## Figures and Tables

**Figure 1 ijms-22-07525-f001:**
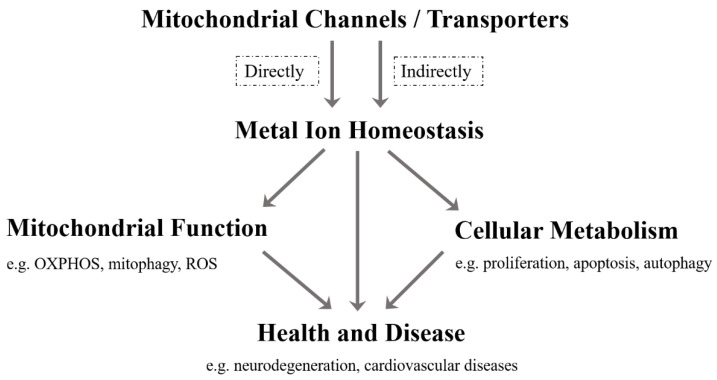
Mitochondrial channels/transporters transport various kinds of metal ions, which directly or indirectly regulates the homeostasis of metal ions. The metal ion homeostasis is essential for mitochondrial function, cellular metabolism, health and disease.

**Figure 2 ijms-22-07525-f002:**
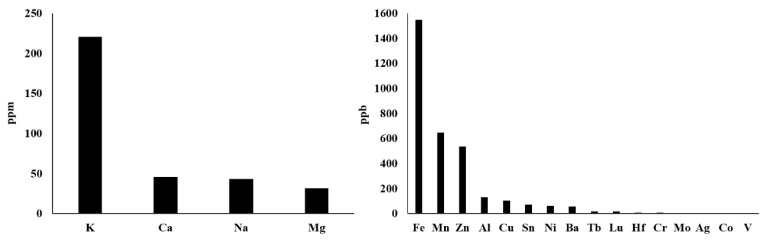
The contents of metal ions in mitochondria of HepG2 cells. The abundance decreases from left to right. HepG2 cells overexpressed the HA tag located on the MOM, and then mitochondria were quickly isolated using a HA-based magnetic bead (HA antibody-conjugated beads) system. After mitochondria were lysed, the abundance of metal ions was measured by ICP–MS.

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
