# Peer review of "Mitochondrial Metal Ion Transport in Cell Metabolism and Disease"

_ijms, 2021, doi:10.3390/ijms22147525_

Round 1

Reviewer 1 Report

The article entitled “Mitochondrial Metal Ion Transport in Cell Metabolism 2 and Disease” presented by Xuan Wang, Peng An, Zhenglong Gu, Yongting Luo and Junjie Luo is quite interesting and may have been of huge interest but it failed within a long list of review on mitochondrial ion transport review articles.

To me, it looks like a catalog with a listing of all the major metal ions, one by one. The authors may go more in deep in their project and develop some part that are underestimated like metal ion and apoptosis, metal ion and autophagy.

A paragraph about potential chelators would have been useful.

An exhaustive abbreviation paragraph is also needed;

But at whole, I do not find that this article is of sufficient quality to be published.

Author Response

The article entitled “Mitochondrial Metal Ion Transport in Cell Metabolism 2 and Disease” presented by Xuan Wang, Peng An, Zhenglong Gu, Yongting Luo and Junjie Luo is quite interesting and may have been of huge interest but it failed within a long list of review on mitochondrial ion transport review articles.

Response: We sincerely thank the reviewer for insightful suggestions that allow us to improve our manuscript. Although there are many review articles on mitochondrial ion transport, all these articles focus on very limited types of mitochondrial metal ions, and most of them are limited to talking about the transport of mitochondrial Ca2+ and K+. There is a lack of systematic reviews on the transport of other mitochondrial metal ions and their relationship with cell metabolism and disease. Our review is the first to introduce the transport of so many kinds of mitochondrial metal ions in cell metabolism and disease. In addition, our review is also an update of the latest research progress.

To me, it looks like a catalog with a listing of all the major metal ions, one by one. The authors may go more in deep in their project and develop some part that are underestimated like metal ion and apoptosis, metal ion and autophagy.

Response: Thank you for suggestions. In our manuscript, we have introduced all the major mitochondrial metal ions transport channels/transporters, and clarify whether they are closely related to cell metabolism and diseases. Here, we emphasize the metal ions in mitochondria. If there is indeed relevant research to support, we additionally discuss mitochondrial metal ion and apoptosis, mitochondrial metal ion and autophagy in each metal ion section.

Mitochondrial metal ion and apoptosis: For example, “Excessive Ca2+ within mitochondria can induce apoptosis by mPTP.” “The reduced mitochondrial Ca2+ overload under pro-apoptotic stimuli significantly alleviated the apoptotic response.” “MitoKATP channels have potential effects on multiple pathological processes, such as improving cardiac function, inhibiting apoptosis in diabetic cardiomyopathy.” “Another K+ influx channel, mitoTASK-3, whose function may be involvement in ROS production and apoptosis, needs to be further understood.” “Mg2+ participates in the release of Cytochrome C, and the concentration of Mg2+ in the cytoplasm of apoptotic cells is also significantly increased in vitro. But in human colon cancer cells, intracellular Mg2+ content decreases during mitochondria-mediated apoptosis. Regardless of this controversy, it is certain that there is a close relationship between Mg2+ homeostasis and apoptosis. In short, the effects of Mg2+ on mitochondrial functions mainly focus on energy metabolism, mitochondrial Ca2+ processing and apoptosis.” “The MRS2 channel seems to be involved in apoptosis, and its expression is strictly lined to Mg2+ content inside the cells and mitochondria. In fact, in vitro experiments conducted on human cells of conditional knockdown and overexpression of MRS2 channels induce cell death or lead to a minor susceptibility to apoptotic pharmacological insults. Moreover, mitochondria mediated apoptosis is linked to multidrug-resistant (MDR) phenotype and gastric cancer, and the function of human MRS2 protein may be a promising target for MDR reversal therapy.”

Mitochondrial metal ion and autophagy: For instance, “A study has indicated that crosstalk between mitochondrial Ca2+ uptake and autophagy in skeletal muscle.” “The defect of KHE can lead to increased matrix K+ content, swelling, and attenuation of autophagy.” “An iron loss induced by iron chelator triggers mitophagy.”

A paragraph about potential chelators would have been useful.

Response: We thank the reviewer for this suggestion and we have added a paragraph about potential chelators in “10. Concluding remarks and prospects” section.

Actually, within mitochondria there may be completely another concentrations of metal ions (e.g. calcium) than in cytoplasm which is crucial for proper functioning of mitochondria as an organelle. Mitochondrial metal ions homeostasis plays a very important role in health. Excessive metal ions mean dyshomeostasis, which affects cell metabolism and disease. Metal ions chelators have been reported to solve excess metal ions in cells, and get a certain range of applications. However, metal ions chelators applied to mitochondria are rare at present. Whether chelators acting on the cytoplasm can also regulate the homeostasis of mitochondrial metal ions remains to be further studied. The development of potential chelators that act on mitochondrial metal ions will be challenging and of great significance.

An exhaustive abbreviation paragraph is also needed.

Response: Thanks and we have added an exhaustive abbreviation paragraph.

Abbreviations: OXPHOS, oxidative phosphorylation; ATP, adenosine triphosphate; ROS, reactive oxygen species; MIM, mitochondrial inner membrane; MOM, mitochondrial outer membrane; T2D, type 2 diabetes; VDAC, voltage-dependent anion-selective channel; MCU, mitochondrial calcium uniporter; mRYR, mitochondrial ryanodine receptor; ETC, electron-transport chain; mPTP, mitochondrial permeability transition pore; mitoKATP, mitochondrial ATP-sensitive K+ channel; KCa, Ca2+-activated K+ channel; Kv, voltage-gated K+ channel; mitoTASK-3, TWIK-related acid-sensitive K+ channel 3; KHE, K+/H+ exchanger; NHE, Na+/H+ exchanger; NCLX, Na+/Ca2+ exchanger; MRS2, mitochondrial RNA splicing protein 2; Tf/TfR2, transferrin/transferrin receptor 2; SOD, superoxide dismutase; ICP-MS, inductively coupled plasma mass spectrometry.

But at whole, I do not find that this article is of sufficient quality to be published.

Response: We have made corresponding changes based on the valuable comments of the reviewer, and hope that the revised manuscript will be of sufficient quality for publication.

Reviewer 2 Report

The manuscript entitled „Mitochondrial Metal Ion Transport in Cell Metabolism and Disease” addresses the very interesting problem of ionic transport via mitochondrial channels, which is not so broadly scientifically exploited as the transport of ions by the channel proteins in plasma membrane. The submitted review raises the question about the importance of the involvement of the mito-channels in many important physiological processes which are crucial in health and disease.

It is generally well-written and reasonably structured.

I don’t see any citations concerning the issues raised by Authors that are evidently missing.

My comments and suggestions:

  1. In table 1 the appropriate references could be added.
  2. Figure 1 is clear and simple, however it could be improved to be more informative (e.g. some examples could be added)
  3. The sentence: “The content of metal ions in mitochondria is not the more the better” sounds naïve. I think that it would be better to emphasize the fact that within mitochondria there are completely another concentrations of ions (e.g. calcium) than in cytoplasm which is crucial for proper functioning of mitochondria as an organelle.

Author Response

The manuscript entitled “Mitochondrial Metal Ion Transport in Cell Metabolism and Disease” addresses the very interesting problem of ionic transport via mitochondrial channels, which is not so broadly scientifically exploited as the transport of ions by the channel proteins in plasma membrane. The submitted review raises the question about the importance of the involvement of the mito-channels in many important physiological processes which are crucial in health and disease.

It is generally well-written and reasonably structured.

I don’t see any citations concerning the issues raised by Authors that are evidently missing.

Response: We sincerely thank the reviewer for these encouraging and insightful comments.

My comments and suggestions:

In table 1 the appropriate references could be added.

Response: Thanks and we have added references in table 1.

Figure 1 is clear and simple, however it could be improved to be more informative (e.g. some examples could be added)

Response: We thank the reviewer for this good suggestion and have improved Figure 1 in revised manuscript.

The sentence: “The content of metal ions in mitochondria is not the more the better” sounds naïve. I think that it would be better to emphasize the fact that within mitochondria there are completely another concentrations of ions (e.g. calcium) than in cytoplasm which is crucial for proper functioning of mitochondria as an organelle.

Response: Thank you for pointing out this issue. We have deleted the sentence “The content of metal ions in mitochondria is not the more the better”. According to the reviewer’ suggestion, we have added relevant description to the "10. Concluding remarks and prospects" section of our revised manuscript.

Reviewer 3 Report

The manuscript by Wang and colleagues deal  with the significance of  mitochondrial ion channels transportes in human physiology and disease.

The subject matter of this review represents a topic of interest for scientific community belonging to the area of ion channels and transporters that may be involved in the proper physiology functioning and disease development.
In general the article is well written and fluent and covers the major implications of mitochondria ion channels and transporters in cell metabolism and diseases

In the introduction it is correct to talk about channels and transporters in general, however, when describing the activity of a specific protein it has to be specified more clearly if it acts as channels or transporters as their mechanism of action are completely different.

Moreover, I would appreciate if authors clearly state if the cited articles and experiments are performed in vitro (e.g. on human cells) in vivo (e.g. on animals models or patients) .

Based on this premises, translational aspects of channels/transporters in health and disease should also be stressed.

Chapter 5. Mitochondrial Mg2+ should be expanded considering that:

  • Mg2+, as authors correctly state, is involved in energetic metabolism. But I would like to read more about Mg-ATP complex as the biological functional form of ATP for the cell.
  • The involvement of Mg in the apoptotic process: the authors cite two works were Mg seems to increase during apoptosis. However, other authors report the opposite: Cappadone et al. 2012, Magnes. Res). This reference should be added for completeness and the discussed.
  • MRS2 channels seems to be also involved in apoptosis and that its expression it strictly lined to Mg content inside the cells and mitochondria. In fact, in vitro experiments conducted on human cells of conditional knockdown and overexpression of MRS2 channels induce cell death or lead to a minor susceptibility to apoptotic pharmacological insults. Moreover, mitochondria mediated apoptosis is linked to MDR phenotype and gastric cancer. [Piskacek M et al, Cell Mol Med. 2009; Merolle L,.et al Metallomics. 2018; Chen Y et al, Cancer Biol Ther. 2009 ]
  1. Mitochondrial iron ion:
  • -It is conceivable that mitochondrial iron homeostasis is involved in haematological disease such as thalassemia. Iron metabolism is largely affected by this condition and may lead to iron overload associated side effects. I would appreciate if authors discuss also  this aspect.

Specific comments:

Plesase control the spaces before brackets of the references.

Page 1 line 17 please rephrase and shortening the sentence.

Page 2 line 28 please change the starting sentence, “Mitochondria are cytoplasmic organelles crucial to life”              

Page 2 line 30 change cell with cells

Page 2 line 31 control spacing

Page 2 line 45  “Impaired metal ion homeostasis is linked to mitochondrial dysfunction” please specify if the impairment is at cellular level or at systemic level

Page 2 line 52 Please rephrase, the sentence is misleading

Page 2 line 56 Please delete the sentence, repetition

Page 5 line 200 Cytochrome C

-Caption of figure 1 has to be expanded including the direct and indirect action of metal ion homeostasis orchestrated by channels and transporters on health and disease. Use the plural for channels and transporters.

Author Response

The manuscript by Wang and colleagues deal with the significance of mitochondrial ion channels transporters in human physiology and disease.

The subject matter of this review represents a topic of interest for scientific community belonging to the area of ion channels and transporters that may be involved in the proper physiology functioning and disease development.
In general the article is well written and fluent and covers the major implications of mitochondria ion channels and transporters in cell metabolism and diseases

Response: We sincerely thank the reviewer for the encouraging and insightful comments.

In the introduction it is correct to talk about channels and transporters in general, however, when describing the activity of a specific protein it has to be specified more clearly if it acts as channels or transporters as their mechanism of action are completely different.

Response: We thank the reviewer for pointing out this issue, and we have changed the description more clearly in revised manuscript.

Moreover, I would appreciate if authors clearly state if the cited articles and experiments are performed in vitro (e.g. on human cells) in vivo (e.g. on animal models or patients).

Response: Thank you for this suggestion, and we have improved the related description.

Based on this premises, translational aspects of channels/transporters in health and disease should also be stressed.

Response: Thanks. If there are reports on translational aspects of mitochondrial metal ion channels/transporters in health and disease, we have discussed in each chapter. But in fact, relevant reports are rare and need more attention. Furthermore, we also emphasized translational research in the "10. Concluding remarks and prospects" section.

Chapter 5. Mitochondrial Mg2+ should be expanded considering that:

  • Mg2+, as authors correctly state, is involved in energetic metabolism. But I would like to read more about Mg-ATP complex as the biological functional form of ATP for the cell.

Response: According to the reviewer’s suggestion, we have added more content about Mg-ATP complex in the revised manuscript.

  • The involvement of Mg in the apoptotic process: the authors cite two works were Mg seems to increase during apoptosis. However, other authors report the opposite: Cappadone et al. 2012, Magnes. Res). This reference should be added for completeness and the discussed.

Response: Thank you and we have revised.

  • MRS2 channels seems to be also involved in apoptosis and that its expression it strictly lined to Mg content inside the cells and mitochondria. In fact, in vitro experiments conducted on human cells of conditional knockdown and overexpression of MRS2 channels induce cell death or lead to a minor susceptibility to apoptotic pharmacological insults. Moreover, mitochondria mediated apoptosis is linked to MDR phenotype and gastric cancer. [Piskacek M et al, Cell Mol Med. 2009; Merolle L,.et al Metallomics. 2018; Chen Y et al, Cancer Biol Ther. 2009 ]

Response: Thanks and we have added these content.

     7. Mitochondrial iron ion:

  • -It is conceivable that mitochondrial iron homeostasis is involved in haematological disease such as thalassemia. Iron metabolism is largely affected by this condition and may lead to iron overload associated side effects. I would appreciate if authors discuss also this aspect.

Response: We thank the reviewer for this suggestion and we have added relevant discussion in “7. Mitochondrial iron ion” section.

Specific comments:

Plesase control the spaces before brackets of the references.

Response: Our apology and corrected.

Page 1 line 17 please rephrase and shortening the sentence.

Response: Thanks and we have revised.

Page 2 line 28 please change the starting sentence, “Mitochondria are cytoplasmic organelles crucial to life”  

Response: Thank you and we have changed.

Page 2 line 30 change cell with cells

Response: Our apology and corrected.

Page 2 line 31 control spacing

Response: Our apology and corrected.

Page 2 line 45  “Impaired metal ion homeostasis is linked to mitochondrial dysfunction” please specify if the impairment is at cellular level or at systemic level

Response: Thanks and we have revised.

Page 2 line 52 Please rephrase, the sentence is misleading

Response: Our apology and revised.

Page 2 line 56 Please delete the sentence, repetition

Response: Our apology and corrected.

Page 5 line 200 Cytochrome C

Response: Our apology and corrected.

-Caption of figure 1 has to be expanded including the direct and indirect action of metal ion homeostasis orchestrated by channels and transporters on health and disease. Use the plural for channels and transporters.

Response: Thank you for pointing out these issues. According to the reviewer’ suggestions, we have improved Figure 1 and its caption in our revised manuscript.